# Prevalence and Types of Non-Nutritive Sweeteners in the New Zealand Food Supply, 2013 and 2019

**DOI:** 10.3390/nu13093228

**Published:** 2021-09-16

**Authors:** Rachel Nunn, Leanne Young, Cliona Ni Mhurchu

**Affiliations:** 1Faculty of Medical and Health Sciences, University of Auckland, Private Bag 92019, Auckland 1142, New Zealand; rnun189@aucklanduni.ac.nz; 2National Institute for Health Innovation, University of Auckland, Private Bag 92019, Auckland 1142, New Zealand; c.nimhurchu@auckland.ac.nz; 3The George Institute for Global Health, Level 5, 1 King Street, Sydney, NSW 2042, Australia; 4Department of Medicine, University of New South Wales, Sydney, NSW 2052, Australia

**Keywords:** non-nutritive sweeteners, New Zealand, food supply, packaged foods, sugar-sweetened beverages, sugar

## Abstract

The widely recognized association between high sugar intakes and adverse health outcomes has increased consumer demand for products lower in sugar. This may lead to increased use of other sweeteners by the food industry. The current study investigated the prevalence and types of non-nutritive sweeteners over time (2013–2019) in New Zealand’s packaged food and beverages, overall and between categories. A New Zealand database of packaged foods and beverages was used to investigate the presence of Food Standards Australia New Zealand Code-approved non-nutritive sweeteners (*n* = 12). Products available in 2013 (*n* = 12,153) and 2019 (*n* = 14,645) were compared. Between 2013 and 2019, the prevalence of non-nutritive sweeteners in products increased from 3% to 5%. The most common non-nutritive sweeteners in both years were acesulphame-potassium, sucralose, aspartame, and stevia, which were predominantly found in special foods (breakfast beverages and nutritional supplements), non-alcoholic beverages, dairy products, and confectionery. The prevalence of non-nutritive sweeteners is increasing over time in New Zealand’s packaged foods and beverages and is likely a consequence of consumer demand for lower-sugar products. Ongoing monitoring of the prevalence and type of NNS is important to detect further increases.

## 1. Introduction

Poor diet is a major contributor to non-communicable disease (NCD) risk globally [1] and in New Zealand (NZ) [2]. Subgroups of the population in NZ, including indigenous Māori, Pacific, and those experiencing socio-economic hardship, experience higher rates of NCDs compared to non-Māori, non-Pacific, and less deprived population groups [2]. Specifically, excess sugar consumption is strongly associated with higher body weight [3], dental decay [4], type 2 diabetes [5] and cardiovascular disease risk factors such as raised blood lipids and high blood pressure [6].

Despite government nutrition guideline recommendations to choose and prepare foods and beverages with little or no added sugar [7], sugar intake in NZ exceeds World Health Organization (WHO) recommendations. Estimates of free (includes added sugar plus sugar naturally occurring in honey, syrups and fruit juices and concentrates, WHO definition [8]) and added sugar (sugar added to food during processing, US FDA definition [9,10]) intake show that only 42% and 12% of the population meet the WHO recommendations of 10% and 5% of total energy, respectively [11]. Non-alcoholic beverages and sugar and sweets contribute most of the free sugar in the NZ diet [12].

Public health strategies to reduce sugar overconsumption include addressing factors that influence the purchasing of high sugar foods (restriction of marketing and price promotions); tax on high-sugar beverages; reformulation of the food supply (setting sugar benchmarks and monitoring sugar content and portion size reduction); consumer education campaigns to increase awareness and providing practical steps to reduce sugar intake [13]; and food labelling of added sugar content [14,15]. The food industry’s response to global sugar reduction recommendations has included a reduction in added sugar and the replacement of sugars with non-nutritive and low-calorie sweeteners. Recent World Health Organization guidelines advising a reduction in ‘free’ sugar consumption, in conjunction with greater consumer awareness of the link between excess sugar consumption and poor health, has driven demand for reduced-sugar foods and an increase in the number and variety of products containing non-nutritive sweeteners (NNS) [8,16,17].

NNS are food ingredients that are considerably sweeter than sugar and are added to food in small amounts to replace sugar. Functionally, NNS add sweetness to foods with little or no energy, thereby reducing the sugar and energy content of products. Such foods may or may not be labelled as ‘diet’, ‘sugar-free’ or ‘low-joule/calorie’ and commonly contain one or more NNS in full or partial replacement of sugar [15]. A recent study of packaged food in Hong Kong noted that many foods containing NNS were not labelled ‘diet’ and therefore NNS may be consumed in larger amounts than previously thought [18]. These foods can be useful to people following sugar or energy restricted diets for a medical condition, including diabetes, or other more general diet and lifestyle improvement reasons. In NZ, non-nutritive sweeteners are regulated by the Food Standards Australia New Zealand (FSANZ) Code as food additives and are termed ‘Intense Sweeteners’ [19]. These regulations specify the amounts and types of NNS permitted in foods. Any NNS used as an ingredient in the manufacture of a food product must be listed in the ingredients list on food packaging.

The intake of sweeteners can be assessed by food regulatory authorities at a population level against the Acceptable Daily Intake (ADI) which is the amount of a food additive that can be consumed (per kilogram of body weight) each day over a person’s life without adverse effects [20]. International food regulatory and other health or scientific agencies, including the Food and Agricultural Organization [21], are responsible for the development of ADI levels. A recent review of the intakes of the seven most used NNS globally, based on studies since 2008, showed that overall intakes remained within ADI guidelines, though ongoing monitoring was advised in response to global dietary guidelines to lower sugar intake [22].

Globally, NNS are found in a wide range of products including beverages, dairy products, confectionery, condiments, processed fruit and vegetable products and snack foods [15]. Most of the data on consumption of NNS is from the USA and overall shows an increase in the consumption of NNS over time, with beverages being the predominant source [15]. Beverages, specifically carbonated soft drinks, were the main source of NNS in diets of adults and children in Australia [23]. Non-alcoholic beverages were also the major contributor of NNS in the Spanish diet (36%), followed by sugar and sweets (14.2%) and milk and dairy products (7%) [24].

Internationally, there are few studies investigating the prevalence of NNS in the food supply over time [15]. Only one cross-sectional study has previously reported on the prevalence of NNS in NZ, which was a cross-country comparison showing that 1% of foods contained a NNS in 2016 [25]. Therefore, little is known of the current prevalence of NNS in NZ’s packaged foods and beverages and likewise any trend over time. The aim of this study was to determine the prevalence and types of NNS overall and within food categories in packaged products in NZ in 2013 and 2019.

## 2. Materials and Methods

There are 12 permitted sweeteners in the Food Standards Australia New Zealand (FSANZ) Code [19] (Table 1). Sugar alcohols (e.g., erythritol, sorbitol, mannitol, xylitol) are low-energy (rather than zero energy) food additives that may be used as a sweetener or for other functions in foods but are not classed as intense sweeteners; therefore, they are excluded from this study. The Nutritrack database, an annually updated database of packaged food and non-alcoholic beverages, was the source of data for this study [26]. This database is compiled of rigorous surveys of four supermarket stores in Auckland, representing the two major supermarket retailers operating in New Zealand. Field staff are trained to systematically collect information on product nutrition composition, ingredients and labelling by photographs using a tailored smart-phone application instore. Product details are entered into a secure online database and the accuracy of the data is quality checked in a random sample of 15% of products.

Products for this study were classified as containing a NNS based on keyword searches within the product ingredients. NZ’s food regulations require the word ‘sweetener’ followed by the name or the code number of the sweetener to be listed in the ingredients, hence both options were included in searches (Table 1).

The major food categories in Nutritrack, developed by the Global Food Monitoring Group [27], were used in this analysis. Categories included: Bread and bakery; Cereal and cereal products; Confectionery; Convenience foods; Dairy; Edible oils and oil emulsions, Eggs, Fish and seafood products; Fruit and vegetables; Meat and meat products; Non-alcoholic beverages; Sauces and spreads; Snack foods; Special foods; Sugars, honey, and related products. The Special foods category includes breakfast beverages, diet drink mixes, diet soup mixes, other fitness, or diet products (including table-top sweeteners), protein and diet bars, protein powders, and sports gels. The sub-category of Baby foods and any products without ingredient lists were excluded (*n* = 1030 across both years). Products analyzed totaled 12,153 in 2013 and 14,645 in 2019.

Microsoft Excel (Version 2008 Microsoft 365 MSO) was used to assess prevalence of each NNS overall and by food category. Chi square tests for comparing proportions of NNS between years and food groups were conducted using SPSS (IBM SPSS Statistics 27). A *p*-value of less than 0.05 was considered statistically significant.

## 3. Results

In 2013, 3% all packaged products (*n* = 378) contained at least one NNS. This increased to 5% (*n* = 761) in 2019 (*p* < 0.001) (Table 2). Acesulphame-potassium, sucralose, and aspartame were the most prevalent NNS in 2013 and stevia, sucralose and acesulphame-potassium were the most prevalent in 2019. The prevalence of the nine most common NNS within the major NNS-containing categories in 2013 and 2019 are presented in Figure 1. Data for aspartame-acesulphame and alitame are not shown in Figure 1 as there was only 1 product (in one or two years) and no products containing advantame in either year.

The most common non-nutritive sweeteners in 2013 were acesulphame-potassium (1.1%), sucralose (1%), aspartame (0.95%), and stevia (0.5%) and in 2019, stevia (2.3%), sucralose (1.6%), acesulphame-potassium (1.4%) and aspartame (1.2%) (Table 3). In 2013, no products were identified as containing thaumatin or monk fruit extract and in both years, no products containing advantame were identified. In both 2013 and 2019, approximately one third of products with a NNS as an ingredient contained 2–5 NNS co-occurring in the same product (133 products [35%] in 2013, and 241 products [32%] in 2019). In 2013 and 2019, the predominant food categories containing multiple sweeteners (≥2) were confectionery and non-alcoholic beverages, although the categories of convenience foods, special foods, and sugars, honey and related products contained greater numbers of products with multiple sweeteners compared to 2013. The same three sweeteners, acesulphame-potassium, aspartame, and sucralose, prominently featured in products containing multiple sweeteners, in both years.

The largest proportions of NNS-containing products were found in the special foods category (Table 2). The number and proportion of NNS-containing products in this category rose significantly from 52 products in 2013 (32%) to 118 products (49%) in 2019 (*p* = 0.001). Of the products in 2013, 56% were protein powders, 19% protein and diet bars, and 12% were diet drink mixes. This contrasted with 2019, where the largest proportion was the protein and diet bars subcategory (41%), followed by protein powders (31%) and diet drink mixes (21%). Sucralose remained the most used NNS in the special foods category and increased significantly between years; 35 products (56%) in 2013 and 85 products (63%) in 2019 (*p* = 0.004) (Figure 1).

The second highest NNS-containing category was non-alcoholic beverages. The number and proportion of non-alcoholic beverage products containing a NNS increased significantly between 2013 (175 products, 16%) and 2019 (370 products, 25%) (*p* < 0.001). The subcategory of soft drinks contained approximately 43% and 46% of NNS in 2013 and 2019, respectively. The most used NNS in this category was acesulphame-potassium in 2013 (65 products, 26%) and stevia in 2019 (179 products, 33%).

Within the dairy category, a significant increase was seen in the number of products that contained NNS between 2013 (23 products; 1.4%) and 2019 (58 products; 3%) (*p* = 0.001). In contrast, little change occurred in the overall proportion of products containing NNS in the confectionery category between years although there was a significant increase in the number of products containing stevia (2 products in 2013 to 29 products in 2019) (*p* < 0.001).

## 4. Discussion

Between 2013 and 2019, the use of NNS in NZ’s packaged foods and beverages increased from 3% to 5%. This is higher than the prevalence reported in a previous analysis of NZ (1%) and Australian (1%) products, similar to USA (4%) but not as high as Mexico (11%) in 2016 [25]. The lower prevalence of NNS reported in this study may reflect non-inclusion of the ‘special foods’ category. The prevalence of NNS in NZ in 2019 is similar to the findings of a recent study in Hong Kong which showed 4.5% of packaged foods contained at least 1 NNS [18].

Our study is one of only a few to investigate prevalence of NNS in the food supply over time across a wide variety of packaged food product categories. It uses the Nutritrack database which is annually updated data from the major supermarket stores in New Zealand, estimated to represent 75% of available packaged foods [28]; hence, it provides comprehensive data on food availability and ingredients. One limitation is that our study does not assess purchases, sales, or consumption data therefore results may not reflect actual purchasing or consumption patterns for these products. However, such studies from the USA between 2002 and 2018 [29,30], and Norway [31] provide evidence that consumption and purchases of products containing NNS are increasing.

Use of NNS in NZ is currently concentrated in two food categories: special foods and non-alcoholic beverages. The increasing proportion of NNS-containing products in special foods products may be driven by the growing demand for protein-rich, and low, or no added sugar nutritional supplements. Within the beverages category, almost half of the soft drinks sub-category contained an NNS, similar to a previous study [25] and likely reflects consumer demand and industry response to calls for lower-sugar drink options [25,32]. An international review found similar predominant categories i.e., beverages (soft drinks and juices) as the major source of NNS followed by dairy products, confectionery, and table-top sweeteners (included in our Special Foods category) [33]. Rising use of NNS in non-alcoholic drinks in our study was also reported in a recent study in Slovenia where the use of low and no-calorie sweeteners increased from 13.2% to 15.5% over a two year period (2017 and 2019) [34].

In 2019, the most common NNS used in NZ’s packaged food and beverage products were stevia, sucralose and acesulphame potassium. The predominance of specific NNS within different categories reflects their biochemical and functional properties including relative sweetness with regard to how they are able to successfully alter the taste of the product without affecting consumer acceptability [35]. With regard to changes in the use of specific NNS over time, the decline in aspartame likely results from public concern regarding possible adverse effects [33]. Fluctuations in other NNS may reflect changes in consumer demand as well as availability of new sweeteners. For instance, consumers have reported an avoidance of ‘artificial’ sweeteners in favor of products that are ‘natural’ [30]. This may explain the observed greater use by industry of stevia, which derives from the leaves of the stevia plant. Similarly, the use of monk fruit extract, a naturally occurring sweetener, has also increased since 2013. In contrast, stevia was not the most prevalent NNS in a Spanish study, where acesulfame-potassium, sucralose and aspartame were the main NNS used [24]. The proportion of household purchases of aspartame-containing products declined in the US between 2002 and 2018, while stevia and sucralose-sweetened products increased although aspartame remained the most purchased by volume per capita [29].

There have been conflicting reports on the effect of NNS consumption on health outcomes. Expected positive health benefits of the inclusion of NNS in the diet relate to less sugar and energy content, however, reported effects on health-related outcomes have been conflicting [36]. A recent systematic review of 56 studies among healthy adults showed a limited positive effect of NNS on body weight and fasting blood glucose, but study numbers were small and certainty of evidence was rated as low [36]. No independent effect was found on adults trying to lose weight, glycemic control, blood pressure, kidney disease, cardiovascular disease, and cancer outcomes [36]. A range of different sweeteners were included in this review. Overall, the poor quality of studies in this review led the authors to conclude that there was an absence of evidence for health advantages or harm from NNS intake and more robust studies of longer duration are thus required [36].

The food industry is responding to consumer demand for lower-sugar products with greater use of NNS; however, the industry should be encouraged to develop new products with lower sugar content and reformulate existing products using the successful sodium reduction model [37] by gradually reducing sugar levels over time to improve the consumer’s acceptance of lower sugar, less sweet products [34]. The UK government-led sodium reduction program uses timebound voluntary sodium reduction targets for over 80 processed food categories, which are lowered progressively over time, to prompt food product reformulation and allow time for consumers to adjust to any flavor changes [38]. Specific food category reductions reported were between 13% (crisps) and 49% (branded breakfast cereal) [38]. A similar program has been developed for sugar reduction with time-bound, voluntary 5% and 20% reduction food category targets and has resulted in an overall 3% reduction in sugar content of branded and private label products between 2015 (baseline) and 2019 although it has had little effect on the food eaten away from home [39]. Greater reductions were seen in specific categories of supermarket foods, e.g., breakfast cereals showed a reduction of 13.3%, and yoghurt and fromage frais 13.3%. Government-led reformulation appears to be a promising strategy that supports gradual sugar (and salt) reduction over time.

This study provides an up-to-date overview of the prevalence and types of NNS in packaged foods available in NZ. This benchmark measure of NNS prevalence in the food supply could prove useful to compare the effects of any new policies to reduce sugar consumption which may promote reformulation. The increasing presence of NNS in the food supply and the changing profile of predominant NNS over time indicate the importance of ongoing monitoring of the prevalence and type of NNS. The increasing use of NNS in the food supply in NZ accords with similar trends occurring in other developed countries. Future studies investigating household purchasing and consumption of NNS would aid an understanding of how NNS are impacting the dietary intakes of NZ’s population. Government-led reformulation programs with clear targets to reduce sugar levels in foods and drinks could assist in reducing population sugar intakes.

## Figures and Tables

**Figure 1 nutrients-13-03228-f001:**
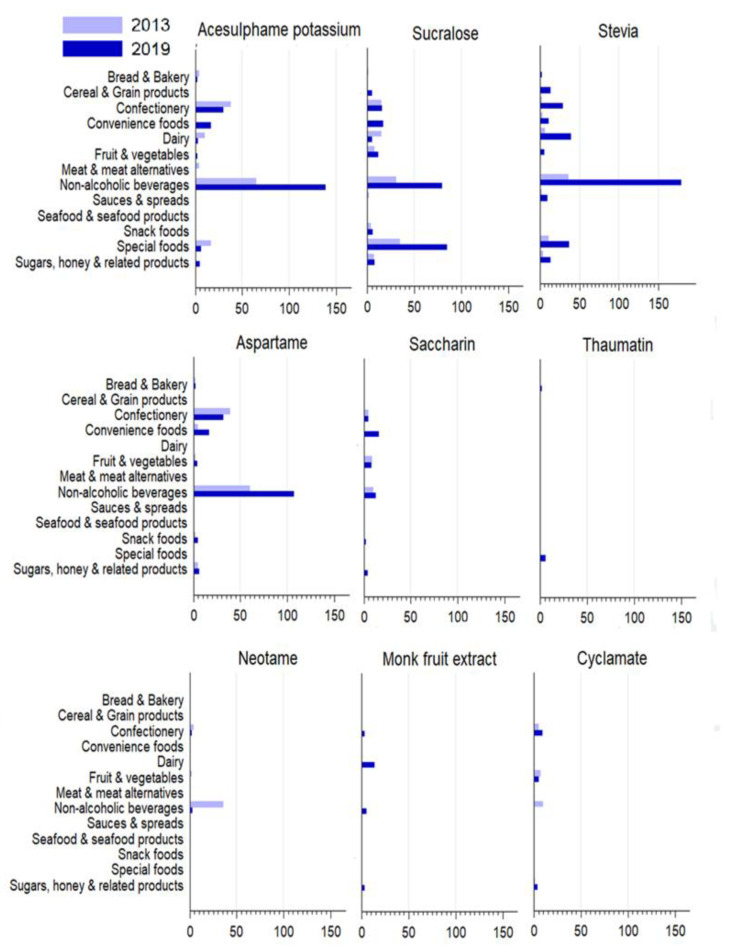
Number of the nine most used NNS in New Zealand packaged foods by major food categories (Categories, Eggs and Edible oils and emulsions, have been excluded).

**Table 1 nutrients-13-03228-t001:** Non-nutritive sweeteners (NNS) and search terms.

Non-Nutritive Sweeteners	Search Terms
Acesulphame potassium	950, acesulphame potassium, acesulphame K, acesulfame potassium, acesulfame K
Advantame	969, advantame
Alitame	956, alitame
Aspartame	951, aspartame
Aspartame-acesulphame salt	962, aspartame-acesulphame salt, aspartame-acesulphame, aspartame-acesulfame salt, aspartame-acesulfame
Cyclamate	952, cyclamate, cyclamic acid, calcium cyclamate, sodium cyclamate
Monk fruit extract	monk fruit extract, monk fruit concentrate, luo han guo extract
Neotame	961, neotame
Saccharin	954, saccharin, calcium saccharine, sodium saccharine, potassium saccharine
Stevia (steviol glycosides)	960, steviol glycoside, stevia extract, stevia, rebaudioside
Sucralose	955, sucralose
Thaumatin	957, thaumatin

**Table 2 nutrients-13-03228-t002:** Number and percentage of products containing at least one NNS by food category and overall, in 2013 and 2019.

	2013		2019		
Food Category	Number in Category	Number and Percentage of Products Containing at Least One NNS	Number in Category	Number and Percentage of Products Containing at Least One NNS	*p* Value
Bread & bakery	1499	7 (0.5)	1679	8 (0.5)	1.0
Cereal & cereal products	1201	1 (0.1)	1485	18 (1.2)	<0.001 *
Confectionery	689	59 (8.6)	906	82 (9.1)	0.79
Dairy	1636	23 (1.4)	1928	58 (3)	0.001 *
Edible oils and emulsions	264	0	320	0	0
Eggs	64	0	86	0	0
Convenience foods	578	8 (1.4)	693	24 (3.5)	0.019 *
Fruit & vegetables	1570	19 (1.2)	1830	28 (1.5)	0.464
Fish & seafood products	441	0	414	1 (0.2)	0.484
Meat & meat products	778	4 (0.5)	953	1 (0.1)	0.181
Non-alcoholic beverages	1104	175 (15.9)	1505	370 (24.6)	<0.001 *
Sauces & spreads	1521	6 (0.4)	1757	10 (0.6)	0.617
Special foods	161	52 (32.3)	243	118 (48.6)	0.001 *
Snack foods	394	6 (1.5)	547	13 (2.4)	0.482
Sugars, honey, and related products	253	18 (7.1)	299	30 (10)	0.289
Total	12,153	378	14,645	761	
Overall prevalence		3.1		5.2	

* Significance at *p* < 0.05.

**Table 3 nutrients-13-03228-t003:** Number and percentage of the total number of products containing each NNS in 2013 and 2019 *.

	2013		2019	
Non-Nutritive Sweeteners	Number of Products Containing an NNS	Percentage of All Products (*n* = 12,153)	Number of Products Containing an NNS	Percentage of All Products (*n* = 14,645)
Stevia	66	0.5	337	2.3
Sucralose	118	1	234	1.6
Acesulphame-potassium	139	1.1	206	1.4
Aspartame	116	0.95	174	1.2
Saccharin	27	0.2	49	0.3
Cyclamate	23	0.2	28	0.2
Monk fruit extract	0	0	25	0.2
Thaumatin	0	0	8	0.1
Neotame	43	0.4	5	0
Aspartame-acesulphame salt	1	0	1	0
Advantame	0	0	0	0
Alitame	1	0	0	0

* Some products contain multiple sweeteners.

## Data Availability

Because of the commercial and legal restrictions to the use of copyrighted material it is not possible to share data openly which reveal the product or company names but unredacted versions of the dataset are available with a licensed agreement that they will be restricted to non-commercial use. For access to Nutritrack, please contact the National Institute for Health Innovation at the University of Auckland at enquiries@nihi.auckland.ac.nz. Nutritrack data for this study was accessed 27 November 2019.

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
