# Peer review of "Prevalence and Types of Non-Nutritive Sweeteners in the New Zealand Food Supply, 2013 and 2019"

_nutrients, 2021, doi:10.3390/nu13093228_

Round 1

Reviewer 1 Report

This is a well written and clearly set out paper, focused on a comparative analysis of the prevalence of non-nutritive sweeteners in the New Zealand food supply between 2013 and 2019.

Some minor comments for consideration are as follows:

Abstract:

Lines 23 and 24: The concluding two sentences of the abstract are valid points; however, they don’t seem to reflect the main discussion of the paper. Additionally, the last sentence implies that sweetener use should be limited/reduced (in a way implying that NNS use is bad), this doesn’t seem to be the focus of the paper or the key conclusion? Perhaps it could be clarified?  

Introduction:

Line 40: It could be helpful to define what is meant by free sugars and added sugars.

Materials and methods:

Line 90: Refer to table 1 or list the 12 included sweeteners before going onto the sugar alcohols that were excluded.

Line 94: I realise it is referenced, however, it could be helpful to have more information on the Nutritrack database in the methods (i.e. is it supermarkets that are surveyed? If so, how many supermarkets are surveyed, are there standardised protocols, and so on).

Lines 107 – 113: Suggest describing the “special foods” category before going onto what is excluded

Results:

Table 2 – Would be helpful to include the years in the table (top row) so that it is very clear that it is a comparison between 2013 and 2019 (like was done for table 3)

Lines 140 – 141: the results on products including multiple NNS is interesting – was this in the same food categories as well as being the same proportion between years? Also, was it a similar mix of NNS? (i.e. was there a standard “set” of NNS that are used?) it could be interesting to add this information to the text and/or table 2.

Figure 1 – Is it possible to make the figure clearer/bigger? At the moment it is quite difficult to interpret on its own, as the labelling on the y axis is quite cramped making it hard to know which bars relate to the food categories.  

Line 158 – There is the word “Turquoise” which seems a bit random and I assume is an error?

Discussion:

Lines 192 – 193 – “label information rather than analysis” – please clarify what is meant here (I am assuming it is chemical analysis of the foods)

General comments on the discussion:

There doesn’t seem to be a clear conclusion, as new information/referenced information is introduced right up until the second to last sentence of the discussion. It could be helpful to have more of a discussion on policy implications (including the comparison to salt reduction efforts) as its own paragraph, and then move onto a stronger concluding statement/paragraph (perhaps this would also help with linking it into the abstract conclusion).

Reviewer 2 Report

The article addresses a very important problem of the use of sweeteners in food and fits perfectly into the current nutritional trends. It is written correctly and shows this problem well in New Zealand over the years. Reducing calories while maintaining a sweet taste are the most important benefits of intense sweeteners. Thanks to their properties, they can be helpful in the prevention and treatment of obesity. The benefits of using these substances in food encourage producers to use them in an ever wider range of products. The availability of an increasing number of products containing intense sweeteners in their composition can simultaneously increase their consumption. This, in turn, encourages them to carry out constant checks on their safety and subject them to subsequent inspections resulting from new conditions of their use. I only have one comment. Maybe the article should mention the ADI of these substances, or Acceptable Daily Intake. The final conclusion, dedicated to producers, about the need to reduce the level of sweetness in many products is very important.
